# Association of immunoglobulin G N-glycosylation with carotid atherosclerotic plaque phenotypes and actual clinical cardiovascular events: a study protocol for a longitudinal prospective cohort study

Cuihong Tian ⓘ ,[1,2] Gehendra Mahara,[3] Hongxia Zhang,[4] Xuerui Tan[1,3]

For numbered affiliations see end of article.

**Correspondence to**
Dr Xuerui Tan;
doctortxr@126.com

## ABSTRACT

**Introduction** Immune-inflammatory response plays a key role in the pathogenesis of atherosclerosis. IgG N-glycosylation is reported to be associated with the 10-year atherosclerotic cardiovascular disease risk score and subclinical atherosclerosis. However, the relationship of IgG glycosylation with actual clinical cardiovascular disease (CVD) events and plaque phenotypes has rarely been investigated. Therefore, this study aims to understand whether IgG glycosylation traits are correlated with actual clinical CVD events and plaque phenotypes.

**Methods and analysis** Designed to verify the efficacy of IgG glycosylation as a risk for CVD events and screen potential biomarkers of CVD to prevent atherosclerosis occurrence, this longitudinal prospective cohort study will be conducted at the First Affiliated Hospital of Shantou University Medical College, China. In total, 2720 participants routinely examined by carotid ultrasound will be divided into different groups according to plaque phenotype characteristics. Ultra-performance liquid chromatography will be performed to separate and detect IgG N-glycans in serum collected at baseline and at the end of the first, second and third years. The primary outcome is the actual clinical CVD composite events, including non-fatal myocardial infarction, death due to coronary heart disease, and fatal or non-fatal stroke.

**Ethics and dissemination** The Clinical Ethics Committee of the First Affiliated Hospital of Shantou University Medical College approved this study (number: B-2021-127). Findings of this study will be submitted for publication in peer-reviewed journals.

**Trial registration number** ChiCTR2100048740.

## INTRODUCTION
### Background and scope

Atherosclerosis (AS) is a cardiovascular disease (CVD) with high morbidity and mortality, accounting for approximately 20% of all deaths worldwide.[1] Ischaemic heart disease (IHD), ischaemic stroke and peripheral vascular disease are the main serious outcomes caused by AS.[2] However, the pathogenesis of

## STRENGTHS AND LIMITATIONS OF THIS STUDY

⇒ This study is a longitudinal prospective cohort study which will indicate change in IgG glycans in people 50–65 years of age.
⇒ The feasibility of this study is high as it does not require treatment of participants with drugs or surgery.
⇒ Due to the protective effects of oestrogen on atherosclerosis, this study will only involve postmenopausal women.
⇒ The cohort is derived from a single centre and represents only one ethnic group, Han Chinese, which will lack ethnic diversity, and the results might not be applicable to the general population.
⇒ Some biases, such as loss to follow-up and other uncontrollable factors, may exist, although this study is designed to control the biases.

AS remains obscure. The main risk factors for AS are family history, hypertension, smoking, dyslipidaemia, diabetes and obesity. Other factors such as metabolism disorders, platelet activation, thrombosis, intimal damage, inflammatory response, oxidative stress and vascular smooth muscle cell activation jointly promote the occurrence and progression of AS.[3–5] In addition, whether some novel factors, such as suboptimal health status and telomere length in peripheral blood leucocytes, could increase the risk of atherosclerotic cardiovascular disease (ASCVD) is being investigated.[6–8]

Studies show that immune inflammation exerts a significant role in atherogenesis.[9] Animal experiments indicate that the initiation, growth, differentiation and rupture of atherosclerotic plaques are influenced by the immune system.[10] A randomised controlled trial involving 10061 patients demonstrated that targeted immune anti-inflammatory pathway therapy

significantly reduces CVD events, independent of lowering lipid levels.[11] Furthermore, germinal centre-derived IgG is a determinant of plaque size and stability and effectively maintains the molecular properties of the aorta.[12] Therefore, the interaction of the adaptive immune system modulates the progression of AS.

Glycosylation is the most abundant and diverse form of post-transcriptional modification in organisms, involved in almost every physiological process, including immune responses.[13 14] Alterations in protein glycosylation are considered as the major event in the transition from health to disease status.[14] It has been demonstrated that IgG glycans are associated with a variety of diseases, such as immune system diseases,[15] cancers,[16–18] nervous system disease,[19] infectious diseases[20 21] and haematological system diseases.[22] Driven by specific enzymes, the complex carbohydrates attached to immunoglobulins have specific regulatory effects, which lead to differences in immune function.[23] A study revealed that glycosylation changes, especially for the most abundant IgG in circulation, directly affect its inflammatory properties.[24] For example, the anti-inflammatory and proinflammatory activity of IgG are determined by sialylation and agalactosylation, respectively.[25 26] Furthermore, a cross-sectional study indicated that deficiency in galactose and sialic acid correlates with blood lipid levels.[27] As the major elements of life, the communication between glycans and lipids even be proposed as a paracentral dogma.[28] Therefore, considering the significant role of inflammation, especially immune inflammation in CVD development, different IgG glycosylation traits may stratify the risk phenotypes for CVD development.

Phenotypes of plaque include with or without plaque, plaque stability, size, location and number. A cross-sectional study based on two independent cohorts reported that the glycosylation traits of IgG are associated with the 10-year ASCVD risk score, subclinical AS and risk factors for CVD, such as smoking, high-density lipoprotein cholesterol and type 2 diabetes mellitus.[29] However, it is unclear whether IgG glycosylation correlates with actual clinical CVD events and plaque phenotypes. Thus, we hypothesise that IgG N-glycan profiling is an independent prognostic indicator of actual clinical CVD composite events.

## Objectives
We aim to understand whether IgG glycosylation traits are correlated with plaque phenotypes and actual clinical CVD composite events, including non-fatal myocardial infarction, coronary heart disease death, and fatal or non-fatal stroke.

## METHODS AND ANALYSIS
### Design and setting
This study is an observational, single-centre longitudinal prospective cohort study to be conducted at the First Affiliated Hospital of Shantou University Medical College (SUMC), Guangdong, China. The protocol applies the checklists of the Standard Protocol Items: Recommendations

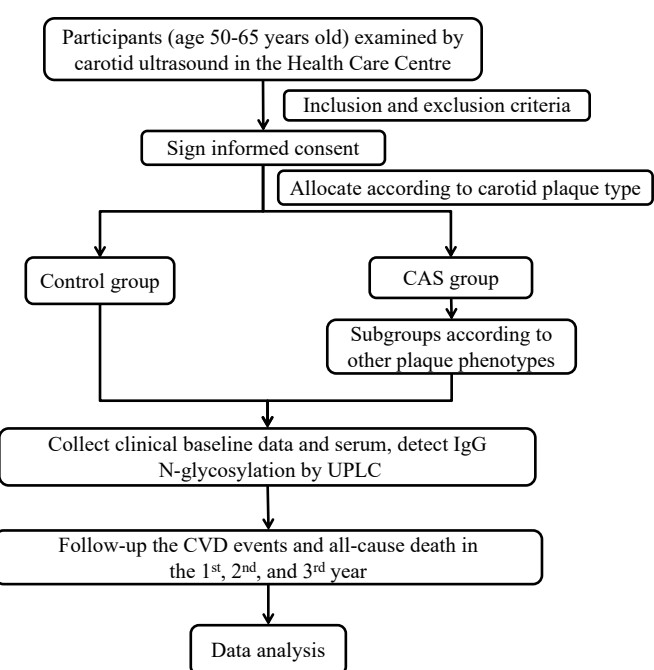

**Figure 1** Participant distribution in the study. CAS, carotid atherosclerosis; CVD, cardiovascular disease; UPLC, ultrahigh-pressure liquid chromatography.

for Interventional Trials (SPIRIT) 2013 statement,[30] a guideline for clinical trial design.

### Participants
This study will be conducted in the Health Care Centre of the First Affiliated Hospital of SUMC, where people over 50 years old are routinely recommended to be examined by carotid ultrasonography for early diagnosis and detection of carotid atherosclerosis (CAS) and early prevention of CVD events. After the clinical study was registered on 15 July 2021, participants conventionally undergoing carotid artery ultrasound examination will be recruited and divided into different groups according to carotid plaque phenotypes (figure 1). Potential eligible participants will be recorded in real time by a sonographer. The research team consists of physicians familiar with ultrasound examination and academic staff skilled in clinical study design and statistical analysis. Furthermore, all investigators will obtain the required research-related training before the start of the study.

### Inclusion criteria
Prospective participants will be carefully evaluated based on the following inclusion criteria:
▶ Age between 50 and 65 years old.
▶ Had carotid ultrasound performed.
▶ Participants or their families provided informed consent.

### Exclusion criteria
Participants with any of the following will not be enrolled in the study:

- ► Participants with complications such as IHD, stroke, infection, immune system disease, nervous system disease, haematological system disease and cancer.
- ► History of lipid-lowering therapy.
- ► History of anticoagulant therapy or antiplatelet therapy or vasodilator therapy.
- ► History of intravenous immunoglobulin or blood transfusion within 1 month.
- ► Irregular follow-up.

## Withdrawal

- ► Participants may withdraw their participation from the study for any reason.
- ► Other circumstances: based on the investigator's judgement, the participant no longer meets the study criteria for any reason.

## Blinding

Participants, sonographers, data gatherers and data analysts are not blinded to group assignments, whereas specimen examiners and follow-up staffers are blinded.

## Data collection and management

Information from the enrolled participants, including smoking history, diabetes status and antihypertensive medication, along with low-density lipoprotein (LDL), high-density lipoprotein, triacylglycerol and total cholesterol levels, will be assembled. In addition, a sonographer will inspect the characteristics of carotid plaque, including plaque size, number, site, blood flow, echo, shape and stability. All data will be kept through EpiData (V.3.1; Denmark), an electronic data capture applied for data entry. Furthermore, 1 mL serum will be collected from each participant at different times (enrolment, first year, second year and third year) after being centrifuged at 604 g for 10 min at 277.15 K. Serum will be stored in the freezer at −80°C for biochemistry and glycan analysis.

## Follow-up

The survival status (surviving/deceased) and actual clinical CVD composite events of the participants, including CVD death, myocardial infarction and stroke, will be followed up at the end of the first, second and third years by two trained investigators through telephone, outpatient visit and physical examination. Furthermore, serum samples will be collected every year and used for detection of IgG glycans (figure 2).

## Sample size

It has been reported that the incidence of CVD events is about 6%.[31] According to Liu et al,[32] among 24 IgG glycan peaks (GPs), GP6 is significantly different between people with and without ischaemic stroke (5.00 (4.26, 5.85) vs 4.48 (3.93, 4.35), p=0.028) and therefore can be used to roughly estimate the sample size. Converting median and quartile to mean and SD is necessary for the sample size.[33–35] Taking the incidence of CVD events and this conversion into consideration, 2267 participants will be needed, based on an α of 0.05 on a two-sided test and 90% power, to obtain a statistical difference. An additional one-fifth is required for the total sample size when considering cases of lost to follow-up and refusal to visit. Finally, a total of 2720 participants are required for the study. The sample size was calculated using PASS software (V.11; NCSS, America).

## Carotid ultrasound

Colour Doppler ultrasonography (Siemens Acuson X150) will be used to scan the carotid artery. Participants will be held on a supine position with the head turned to the opposite side and the probe moved from the root of the neck to the head. The common carotid artery, bifurcation, internal carotid artery and external carotid artery are scanned. Based on the diameter, plaque morphology and echo characteristics, the plaques are classified into three types (soft plaques: uniform or uneven images, irregular form, weak and low echo; hard plaques: strong echo, uneven thickening of the tube wall with enhanced echo and sound shadow behind; mixed plaques: uneven echo, unequal intensity, irregular shape, large image range). Hard plaques are defined as stable plaques, while soft and mixed plaques are regarded as unstable plaques.

## Ultrahigh-pressure liquid chromatography

IgG glycosylation will be detected by ultrahigh-pressure liquid chromatography (UPLC) (LC-30AD, Shimadzu). First, IgG will be isolated with a protein G extraction plate. After being dried and exposed to peptide-N-glycosidase F, N-glycan will be released and labelled with 2-aminobenzamide fluorescent dye. Furthermore, the free markers and reducing agents will be removed by hydrophilic chromatography-solid phase extraction. Subsequently, fluorescently-labelled N-glycans will be separated by hydrophilic chromatography on UPLC.[36] Finally, an automatic processing method with a traditional integration algorithm is used for data processing. Each chromatogram is manually corrected to ensure the same integration interval for all samples.

## Outcome measurements

The primary outcome of this study is the composite events of CVD. The secondary outcome is the 3-year mortality by counting all-cause death.

## Statistical and analytical plans

Categorical variables will be expressed by frequency (percentage) and univariate analysis will be performed using the $\chi^2$ test. Data of normal distribution are shown as mean±SD. The t-test and analysis of variance are used for comparison of two groups and multiple groups, respectively. Non-normally distributed data are represented by median (P25, P75). Univariate analysis is performed by non-parametric testing. Interpolation is carried out according to the type of data loss. If the lost data are of numerical type, the average value is used to fill it. However, if the lost data are non-numerical, the value with the highest occurrence frequency in this index is filled in. Logistic regression will be applied to analyse the

| | Study period | | | | | | |
|---|---|---|---|---|---|---|---|
| | Enrollment | Distribution | UPLC | Follow-up | | | |
| Timepoint | D0 | D1 | W1 | Y1 | Y2 | Y3 | Close-out |
| **Enrollment** | | | | | | | |
| Prescreening | X | | | | | | |
| Eligibility screen | X | | | | | | |
| Informed consent | X | | | | | | |
| **Distribution** | | | | | | | |
| Control group | | X | | | | | |
| CAS group | | X | | | | | |
| **Assessments** | | | | | | | |
| Gender | | X | | | | | |
| Age | | X | | | | | |
| Smoking | | X | | | | | |
| Diabetes | | X | | | | | |
| Hypotensor | | X | | | | | |
| LDL | | X | | | | | |
| HDL | | X | | | | | |
| TC | | X | | | | | |
| TG | | X | | | | | |
| **Carotid ultrasound** | | | | | | | |
| Size | | X | | | | | |
| Number | | X | | | | | |
| Situation | | X | | | | | |
| Blood flow | | X | | | | | |
| Echo | | X | | | | | |
| Shape | | X | | | | | |
| Stability | | X | | | | | |
| Serum | | X | | | | | |
| **UPLC** | | | | | | | |
| Ig G glycosylation | | | X | X | X | X | |
| **Outcome** | | | | | | | |
| Non-fatal MI | | | | X | X | X | |
| CHD death | | | | X | X | X | |
| Stroke | | | | X | X | X | |
| All-cause death | | | | X | X | X | |
| Analysis | | | | | | | X |

**Figure 2** Schedule of enrolment, distribution and assessment in the study. CAS, carotid atherosclerosis; CHD, coronary heart disease; D, day; HDL, high-density lipoprotein; LDL, low-density lipoprotein; MI, myocardial infarction; TC, total cholesterol; TG, triacylglycerol; UPLC, ultrahigh-pressure liquid chromatography; W, week; Y, year.

association between IgG N-glycosylation characteristics and CVD events. Subsequently, a glycan risk score will be created to evaluate the associative effects of all glycan traits screened. A p value <0.05 indicates statistical significance. SPSS software (V.26.0) will be used for statistical analysis.

## Monitoring

This study will be conducted critically according to the approved proposal. The data and safety monitoring committee (DMC) of our hospital will undertake the responsibility of data review and interpretation, which is beneficial to the security and privacy of the participants and to the validity and integrity of the data. Six researchers, independent of the sponsor and conflict of interest, make up the DMC, which will evaluate the data after a quarter, a half and three-quarters of the participants voluntarily enrol in this study. The six researchers will be involved in the entire research progress.

## Auditing

Based on state regulations, the audit work will be conducted during the entire study process to trace the changes and guarantee the quality of research data. Representatives from the ethics committee will shoulder the obligation to inform the related parties of all proposal revisions.

## Harms

This non-intervention study is remarkably safe. Participants will get no exposure to any drug-related or surgery-related therapies.

## Amendment

Any modifications to the proposal will be submitted to the Clinical Ethics Committee and DMC for consent.

# ETHICS AND DISSEMINATION
## Ethics approval

The study protocol was approved by the Clinical Ethics Committee of the First Affiliated Hospital of SUMC on 13 July 2021 (number: B-2021-127).

## Informed consent

Before enrolment, the investigator will introduce the main purpose and content of the study to the participants. In addition, participants will be informed of the possible risks and benefits of the research to ensure that their participation is completely voluntary. After participants have signed the informed consent, all study-related data, including personal information, will be kept confidential. This study will be carried out in accordance with the principles of the Declaration of Helsinki.

## Dissemination

All data and findings of this study will be disseminated at scientific conferences or submitted for publication in peer-reviewed journals.

# DISCUSSION

AS can lead to acute myocardial infarction, cerebral infarction and other serious consequences, which seriously affect quality of life and bring a huge economic burden to patients and society. It has been reported that at the beginning of 2010 the total hospitalisation cost of patients with CVD was over ¥40 billion, which contributes to more than 1.60% of the national health expenditure.[37] ASCVD has become a major public health problem worldwide, making the prevention and treatment of CVD urgent.

A large amount of evidence shows that the pathogenesis of AS is mainly related to the adaptive immune-inflammatory response. AS contains IgG antibodies against LDL, oxidised LDL and apolipoprotein B (ApoB).[38] Some studies suggest that IgG antibody titre is positively correlated with AS, whether in mice or humans,[39] and that anti-ApoB IgG antibody can aggravate AS.[40] In vitro experiments demonstrate that IgG and very low-density lipoprotein are responsible for forming arterial lesions.[41] As an immunoinflammatory biomarker, IgG glycosylation is effective in differentiating patients with dyslipidaemia from the healthy, displaying an area under the curve of approximately 0.7 in a cross-sectional study of Chinese Han.[27] Removal of glycans contributes to the disappearance of the anti-inflammatory properties of IgG,[42] while the enrichment of sialic acid is responsible for a 10-fold increase in the anti-inflammatory properties of IgG.[43] Therefore, control of the adaptive immune response through immune regulatory strategies or vaccination is a potential therapeutic direction for AS.

Glycoprotein acetylation (GlycA), a more reliable compound biological marker of systemic inflammation, has potential value in predicting CVD outcomes and reflecting inflammation.[29 44–47] However, GlycA is a glycan signal of a single measure based on nuclear magnetic resonance. Therefore, the association between IgG N-glycosylation with richer glycoprotein traits and CVD composite events will be assessed in this study. If the results of this study show a significant correlation between IgG glycosylated traits and actual clinical CVD composite events, it will provide further evidence on the role of glycosylation modification and immune-inflammatory response in AS, as well as a potential biomarker better than GlycA in predicting CVD events in the clinic. In addition, clarifying the correlation between IgG and AS and revealing the mechanism of atherogenesis are essential to providing new directions for treatment and prevention of CVD. Our study will provide important scientific significance and socioeconomic value for improving the quality of life and decreasing the mortality rate and the socioeconomic burden of patients with CVD.

However, some limitations to this study exist. First, the participants are recruited at a single centre and represent only one ethnic group, Han Chinese, in the Chaoshan area of Guangdong Province and thus lack ethnic diversity. Further, a multicentre cohort study is required for application of results to the general population. Second, follow-up bias is inevitable in this prospective cohort study due to the existence of subjects who may migrate, leave or withdraw from this cohort, although the authors will try their best to control it.

# TRIAL STATUS

Participant recruitment began on 15 July 2021 and will be finished by 31 December 2021.

**Author affiliations**
[1]Department of Cardiovascular Medicine, The First Affiliated Hospital of Shantou University Medical College, Shantou, Guangdong, China
[2]Centre for Precision Health, Edith Cowan University, Perth, Western Australia, Australia
[3]Clinical Research Centre, The First Affiliated Hospital of Shantou University Medical College, Shantou, Guangdong, China
[4]Health Care Centre, The First Affiliated Hospital of Shantou University Medical College, Shantou, Guangdong, China

**Acknowledgements** We would like to thank Dr Stanley Li Lin, who is a native English speaker from America and a foreign teacher at the Department of Cell Biology and Genetics, SUMC, for his helpful comments and English language editing. We are grateful to Xingyu Wang for his helpful advice on the protocol.

**Contributors** CT was responsible for the design and draft of the manuscript. HZ, working as a sonographer, contributed to the collection of clinical information of the participants. GM reviewed this manuscript. XT, acting as a principal investigator, helped to conceive the protocol. The final version of the manuscript was critically reviewed and approved by all authors.

**Funding** This work is supported by the National Natural Science Foundation of China (No. 82073659) and the Provincial Science and Technology Special Fund of Guangdong in 2021 (No. 2021123071-1).

**Competing interests** None declared.

**Patient and public involvement** Patients and/or the public were not involved in the design, or conduct, or reporting, or dissemination plans of this research.

**Patient consent for publication** Not required.

**Provenance and peer review** Not commissioned; externally peer reviewed.

**ORCID iD**
Cuihong Tian http://orcid.org/0000-0002-3688-072X

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
