## [Reviewer comments · BMJ Open]

ARTICLE DETAILS

TITLE (PROVISIONAL)	Association of immunoglobulin G N-glycosylation with carotid atherosclerotic plaque phenotypes and actual clinical cardiovascular events: a study protocol for a longitudinal prospective cohort study
AUTHORS	Tian, Cuihong; Mahara, Gehendra; Zhang, Hongxia; tan, xuerui

VERSION 1 – REVIEW

REVIEWER	Ivan Gudelj Genos Glycoscience Research Laboratory
REVIEW RETURNED	06-Dec-2021

GENERAL COMMENTS	Please, correct or comment the following: 1) English in the article should be corrected. 2)The list of the exclusion criteria must be extended with several others criteria such as: i) IVIG therapy ii) blood transfusion iii) heparin therapy 3) In the article you didn't explain how you are going to control for other factors which are known to influence IgG glycosylation in the same direction as atherosclerosis (e.g. some infectious diseases). Can you please comment that in the article? 4) Also, can you extend the discussion with: i) limitations of the study; ii) the comparison of your study expectations with others related to IgG glycosylation or glycosylation in general in CVD (e.g. GlycA): iii) clinical significance of the study (e.g. how would you implement those results into the clinic)?
---

REVIEWER	Mandy van Hoek Erasmus Medical Center
REVIEW RETURNED	06-Feb-2022

GENERAL COMMENTS	Tian et al have written a study protocol for investigating the associations of IgG glycosylation with carotid artery plaques and cardiovascular events This is a study that is of interest. I do have a few questions 1) The sample size calculation as performed on page 10 takes the difference between a group with and without CAS as a starting point and sample size to find a difference in CVD events between people with and without CAS is calculated. However, the research question
---

	is a different one. I.e. whether glycosylations is associated with plaque phenotypes and CVD events. The calculation should be made to estimate the power there is to find a difference in these endpoints given a certain sample size. 2) If I understand correctly people that are being referred for carotid ultrasound are then grouped into CAS and non-CAS. However, would there not be a selection bias here? People are referred to the carotid ultrasound for a medical reason? Therefore the control group may still contain a selected group of people with a certain health problem? Can the authors elaborate a bit more on the selection of participants and the types of referral questions they contain?
--	--

VERSION 1 – AUTHOR RESPONSE

Reviewer 1 (Dr. Ivan Gudelj):

Comments to the Author:

Please, correct or comment the following:

- 1) English in the article should be corrected.

Response: We would like to thank Dr. Gudelj for carefully reviewing our manuscript. To improve the quality of our manuscript, we have asked Dr. Stanley Li Lin, a native English speaker from America, for help and thanked him in the “Acknowledgments” section (Page 19).

“Acknowledgments We would like to thank Dr. Stanley Li Lin, who is native English speaker from America as a foreign teacher at the Department of Cell Biology and Genetics, SUMC, for his helpful comments and English language editing.”

- 2) The list of the exclusion criteria must be extended with several others criteria such as:

- i) IVIG therapy;
- ii) blood transfusion;
- iii) heparin therapy.

Response: We are very grateful for your suggestion. We have to admit that we did not very comprehensively consider the exclusion criteria. Anticoagulant drugs such as heparin, warfarin and rivaroxaban, antiplatelet drugs such as aspirin and clopidogrel, vasodilators such as nitroglycerin and alprostadil have an effect on preventing thrombosis. Moreover, intravenous immunoglobulin (IVIG) therapy and blood transfusion will influence the blood concentration of IgG, whose half-time is about 20~30 days. Thus, we have added “history of anticoagulant therapy or antiplatelet therapy or vasodilator therapy” and “history of intravenous immunoglobulin (IVIG) or blood transfusion within one month”, as additional exclusion criteria (Page 7).

3) In the article you didn't explain how you are going to control for other factors which are known to influence IgG glycosylation in the same direction as atherosclerosis (e.g., some infectious diseases). Can you please comment that in the article?

Response: We would like to thank you for pointing out this issue. Actually, many factors influence IgG N-glycosylation, including as atherosclerosis. We added the sentence "It has been demonstrated that IgG glycans are associated with a variety of diseases, such as immune system diseases¹³, cancers¹⁴¹⁵¹⁶, nervous system disease¹⁷, infectious diseases¹⁸¹⁹, and hematological system diseases²⁰." in the "Introduction" section (Paragraph 3, Page 4). As a result, this study will try to exclude participants with complications, such as infection, tumors, and immune, blood and nervous system diseases. We have corrected the first exclusion to "Participants with complications such as IHD, stroke, infection, immune system disease, nervous system disease, hematological system disease and cancer" (Page 7).

13. Parekh RB, Dwek RA, Sutton BJ, et al. Association of rheumatoid arthritis and primary osteoarthritis with changes in the glycosylation pattern of total serum IgG. *Nature* 1985;316(6027):452-7. doi: 10.1038/316452a0 [published Online First: 1985/08/01]
14. Yi CH, Weng HL, Zhou FG, et al. Elevated core-fucosylated IgG is a new marker for hepatitis B virus-related hepatocellular carcinoma. *Oncoimmunology* 2015;4(12):e1011503. doi: 10.1080/2162402x.2015.1011503 [published Online First: 2015/11/21]
15. Bones J, Byrne JC, O'Donoghue N, et al. Glycomic and glycoproteomic analysis of serum from patients with stomach cancer reveals potential markers arising from host defense response mechanisms. *J Proteome Res* 2011;10(3):1246-65. doi: 10.1021/pr101036b [published Online First: 2010/12/15]
16. Qian Y, Wang Y, Zhang X, et al. Quantitative analysis of serum IgG galactosylation assists differential diagnosis of ovarian cancer. *J Proteome Res* 2013;12(9):4046-55. doi: 10.1021/pr4003992 [published Online First: 2013/07/17]
17. Costa J, Streich L, Pinto S, et al. Exploring Cerebrospinal Fluid IgG N-Glycosylation as Potential Biomarker for Amyotrophic Lateral Sclerosis. *Mol Neurobiol* 2019;56(8):5729-39. doi: 10.1007/s12035-019-1482-9 [published Online First: 2019/01/24]
18. Giron LB, Azzoni L, Yin X, et al. Hepatitis C virus modulates IgG glycosylation in HIV co-infected antiretroviral therapy suppressed individuals. *Aids* 2020;34(10):1461-66. doi: 10.1097/qad.0000000000002558 [published Online First: 2020/07/18]
19. Šimurina M, de Haan N, Vučković F, et al. Glycosylation of Immunoglobulin G Associates With Clinical Features of Inflammatory Bowel Diseases. *Gastroenterology* 2018;154(5):1320-33.e10. doi: 10.1053/j.gastro.2018.01.002 [published Online First: 2018/01/09]
20. Lauc G, Huffman JE, Pučić M, et al. Loci associated with N-glycosylation of human immunoglobulin G show pleiotropy with autoimmune diseases and haematological cancers. *PLoS Genet* 2013;9(1):e1003225. doi: 10.1371/journal.pgen.1003225 [published Online First: 2013/02/06]

4) Also, can you extend the discussion with:

i) limitations of the study;

Response: Thanks very much for the suggestion. We have extended the limitations of this study as below in the "discussion" section (Paragraph 4, Page 15).

“However, some limitations in this study exist. Firstly, the participants are recruited at a single center and represent only one ethnic group, Han Chinese, in the Chaoshan area of Guangdong Province, and thus lacks ethnic diversity. Further, a multi-center cohort study is required for application of results to the general population. Secondly, follow-up bias is inevitable in this prospective cohort study due to the existence of subjects who may migrate, leave, or withdraw this cohort, although the authors will try their best to control it.”

ii) the comparison of your study expectations with others related to IgG glycosylation or glycosylation in general in CVD (e.g. GlycA):

Response: We would like to thank you for this valuable comment. We have added the comparison of our study expectations with GlycA as below in the “discussion” section (Paragraph 3, Page 14).

“Glycoprotein acetylation (GlycA), a more reliable compound biological marker of systemic inflammation, has potential value in predicting CVD outcomes and reflecting inflammation^{41 42 43 44 26}. However, GlycA is a glycan signal of a single measure based on nuclear magnetic resonance (NMR). Therefore, the association between IgG N-glycosylation with richer glycoprotein traits and CVD composite events will be assessed in this study.”

41. Duprez DA, Otvos J, Sanchez OA, et al. Comparison of the Predictive Value of GlycA and Other Biomarkers of Inflammation for Total Death, Incident Cardiovascular Events, Noncardiovascular and Noncancer Inflammatory-Related Events, and Total Cancer Events. *Clin Chem* 2016;62(7):1020-31. doi: 10.1373/clinchem.2016.255828 [published Online First: 2016/05/14]
42. Akinkuolie AO, Buring JE, Ridker PM, et al. A novel protein glycan biomarker and future cardiovascular disease events. *Journal of the American Heart Association* 2014;3(5):e001221. doi: 10.1161/jaha.114.001221 [published Online First: 2014/09/25]
43. Lawler PR, Akinkuolie AO, Chandler PD, et al. Circulating N-Linked Glycoprotein Acetyls and Longitudinal Mortality Risk. *Circulation research* 2016;118(7):1106-15. doi: 10.1161/circresaha.115.308078 [published Online First: 2016/03/10]
44. McGarrah RW, Kelly JP, Craig DM, et al. A Novel Protein Glycan-Derived Inflammation Biomarker Independently Predicts Cardiovascular Disease and Modifies the Association of HDL Subclasses with Mortality. *Clin Chem* 2017;63(1):288-96. doi: 10.1373/clinchem.2016.261636 [published Online First: 2016/11/05]

iii) clinical significance of the study (e.g. how would you implement those results into the clinic)?

Response: We have extended the clinical significance of this study as below in “discussion” section (Paragraph 3, Pages 14, 15).

“If the results of this study show a significant correlation between IgG glycosylated traits and actual clinical CVD composite events, it will provide further evidence on the role of glycosylation modification and immune-inflammatory response in AS, as well as be a potential biomarker better than GlycA in predicting CVD events in the clinic. In addition, clarifying the correlation between IgG and AS, and revealing the mechanism of atherogenesis are essential for providing new directions for the treatment

and prevention of CVD. Our study will provide important scientific significance and socio-economic value for improving the quality of life, decreasing the mortality rate and the socio-economic burden of CVD patients.”

Thank you so much to Dr. Gudelj for the comments.

Reviewer 2 (Dr. Mandy van Hoek):

Tian et al have written a study protocol for investigating the associations of IgG glycosylation with carotid artery plaques and cardiovascular events. This is a study that is of interest.

Response: We would like to thank Dr. Hoek for the positive assessment after carefully reviewing our manuscript.

I do have a few questions:

1) The sample size calculation as performed on page 10 takes the difference between a group with and without CAS as a starting point and sample size to find a difference in CVD events between people with and without CAS is calculated. However, the research question is a different one. I.e. whether glycosylations is associated with plaque phenotypes and CVD events. The calculation should be made to estimate the power there is to find a difference in these endpoints given a certain sample size.

Response: We would like to thank you for pointing out this issue. We re-calculated the sample size according to your suggestion. Given the incidence of CVD, the difference of IgG N-glycosylation in primary endpoints, and the missing rate, a total of 2720 participants are required for this study. We elaborate on this in the “sample size” section as below (Page 9).

“It has been reported that the incidence of CVD events is about 6%²⁸. According to Liu et al., among 24 glycans of IgG, GP6 is significantly different between people with and without ischemic stroke [5.00(4.26, 5.85) vs. 4.48(3.93, 4.35), $P=0.028$]²⁹, and therefore can be used to roughly estimate the sample size. Converting median and quartile to mean and standard deviation is necessary for the sample size³⁰⁻³². Taking the incidence of CVD events and this conversion into consideration, 2267 participants will be needed, based on an $\alpha=0.05$ on a two-sided test and 90% power, to obtain a statistical difference. An additional one-fifth is required for the total sample size when considering the cases of lost follow-up and refusal to visit. Finally, a total of 2720 participants are required in the study. The sample size was calculated by using PASS software (NCSS, America, version 11).”

28. Kotruchin P, Hoshide S, Kario K. Carotid atherosclerosis and the association between nocturnal blood pressure dipping and cardiovascular events. *J Clin Hypertens (Greenwich)* 2018;20(3):450-55. doi: 10.1111/jch.13218 [published Online First: 2018/02/17]

29. Liu D, Zhao Z, Wang A, et al. Ischemic stroke is associated with the pro-inflammatory potential of N-glycosylated immunoglobulin G. *J Neuroinflammation* 2018;15(1):123. doi: 10.1186/s12974-018-1161-1 [published Online First: 2018/04/28]
30. Shi J, Luo D, Weng H, et al. Optimally estimating the sample standard deviation from the five-number summary. *Res Synth Methods* 2020;11(5):641-54. doi: 10.1002/jrsm.1429 [published Online First: 2020/06/21]
31. Luo D, Wan X, Liu J, et al. Optimally estimating the sample mean from the sample size, median, mid-range, and/or mid-quartile range. *Stat Methods Med Res* 2018;27(6):1785-805. doi: 10.1177/0962280216669183 [published Online First: 2016/09/30]
32. Wan X, Wang W, Liu J, et al. Estimating the sample mean and standard deviation from the sample size, median, range and/or interquartile range. *BMC Med Res Methodol* 2014;14:135. doi: 10.1186/1471-2288-14-135 [published Online First: 2014/12/20]

2) If I understand correctly people that are being referred for carotid ultrasound are then grouped into CAS and non-CAS. However, would there not be a selection bias here? People are referred to the carotid ultrasound for a medical reason? Therefore, the control group may still contain a selected group of people with a certain health problem? Can the authors elaborate a bit more on the selection of participants and the types of referral questions they contain?

Response: We apologize for the confusing description in our first manuscript. For people over 50 years old in the Health Care Center of the First Affiliated Hospital of SUMC, a physical examination center, carotid ultrasonography is routinely recommended for early diagnosis and detection of CAS, and early prevention of CVD events, not for a specific medical reason. Therefore, there is no selection bias and referral here. We elaborate more on the selection of participants to avoid this confusion (Page 6).

“This study will be conducted in the Health Care Center of the First Affiliated Hospital of SUMC, where people over 50 years old are routinely recommended to be examined by carotid ultrasonography for early diagnosis and detection of CAS, and early prevention of CVD events. After the clinical study registered on July 15, 2021, participants conventionally undergoing carotid artery ultrasound examination will be recruited and divided into different groups according to carotid plaque phenotypes.”

VERSION 2 – REVIEW

REVIEWER	Ivan Gudelj Genos Glycoscience Research Laboratory
REVIEW RETURNED	19-May-2022
GENERAL COMMENTS	I recommend this paper for publication.
REVIEWER	Mandy van Hoek Erasmus Medical Center
REVIEW RETURNED	10-May-2022
GENERAL COMMENTS	My comments have been sufficiently answered, I have no further comments